# Efficient Bayesian Computational Imaging with a Surrogate Score-Based Prior

## Abstract

We propose a surrogate function for efficient use of score-based priors for Bayesian inverse imaging. Recent work turned score-based diffusion models into probabilistic priors for solving ill-posed imaging problems by appealing to an ODE-based log-probability function. However, evaluating this function is computationally inefficient and inhibits posterior estimation of high-dimensional images. Our proposed surrogate prior is based on the evidence lower-bound of a score-based diffusion model. We demonstrate the surrogate prior on variational inference for efficient posterior sampling of large images. Compared to the exact prior used in previous work, our surrogate prior accelerates optimization of the variational distribution by at least two orders of magnitude. We also find that our principled approach achieves higher-fidelity image-reconstruction than non-Bayesian baselines that involve hyperparameter-tuning at inference. Our work establishes a practical path forward for using score-based diffusion models as general-purpose priors for computational imaging.

## 1 Introduction

Ill-posed image reconstruction requires a prior to constrain the reconstruction according to desired image statistics. From a Bayesian perspective, the prior influences both the uncertainty and the richness of the estimated image. Although diffusion-based generative models represent rich image priors, leveraging these priors for Bayesian image-reconstruction remains a challenge. True posterior sampling with an unconditional diffusion model is intractable, so most previous methods heavily approximate the posterior [9; 13; 14; 19] or disregard measurement noise [5; 7; 8; 6; 11; 24; 1]. Recent work demonstrated how to turn score-based diffusion models into probabilistic priors (*score-based priors*) for Bayesian imaging [10]. However, this method requires the exact probability of a proposed image to be evaluated with a computationally-expensive ordinary differential equation (ODE), requiring days to a week to reconstruct even a $32 \times 32$ image [10]. We present a method for Bayesian inference with a score-based prior that is both principled and computationally efficient.

Although computing exact probabilities under a diffusion model is inefficient or even intractable, computing the evidence lower-bound [22; 12] is computationally efficient and feasible for high-dimensional images. Thus we propose to use this evidence lower-bound as a surrogate for the exact score-based prior. In particular, we use the evidence lower-bound of a score-based diffusion model [22] as a substitute for the exact log-probability function. This function can be plugged into any inference algorithm that requires the value or gradient of the posterior log-density. When it is used in variational inference, we find at least two orders of magnitude in speedup of optimizing the variational distribution. Furthermore, our approach reduces GPU memory requirements, as there is no need to evaluate and backpropagate through an ODE. These efficiency improvements make it practical to perform inference with score-based priors.

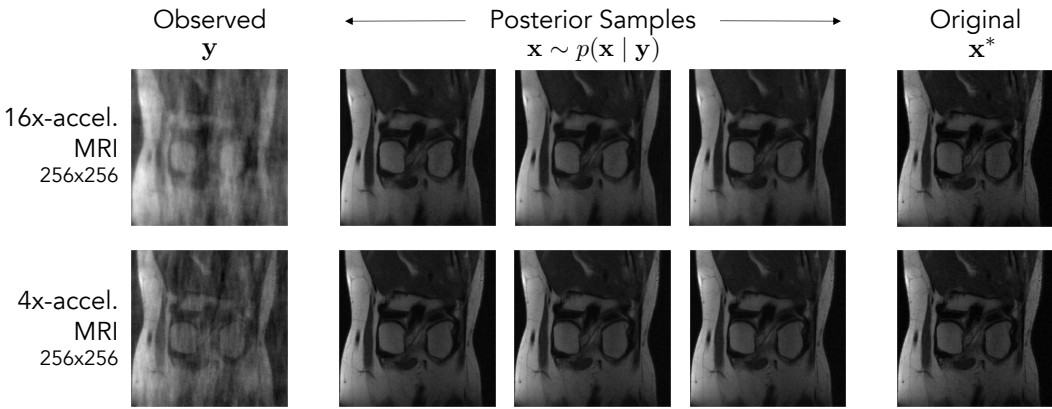

| Observed | ← ——————— Posterior Samples ——————— → | Original |
| $\mathbf{y}$ | $\mathbf{x} \sim p(\mathbf{x} \mid \mathbf{y})$ | $\mathbf{x}^*$ |

Figure 1: High-dimensional Bayesian inference with a surrogate score-based prior. We propose a surrogate prior for efficient use of score-based diffusion models as priors for Bayesian imaging. Here we show posterior samples (estimated with variational inference) for accelerated MRI of $256 \times 256$ knee images with a score-based diffusion-model prior. The first row shows reconstruction from $16\times$-reduced MRI measurements. The second row shows reconstruction given more $\kappa$-space measurements, i.e., $4\times$-reduced MRI. Bayesian imaging at this image resolution is computationally infeasible with the previous ODE-based approach. Our proposed surrogate prior enables efficient yet principled inference with diffusion-model priors, resulting in inferred posteriors where the true image is within three standard deviations of the posterior mean for 96% and 99% of the pixels for $16\times$- and $4\times$-acceleration, respectively.

In this paper, we describe our variational-inference approach to efficiently estimate a posterior with a surrogate score-based prior. We provide experimental results to validate the proposed surrogate prior, including high-dimensional posterior samples of sizes up to $256 \times 256$, a resolution infeasible with the exact prior. In the setting of accelerated MRI, we quantify time- and memory-efficiency improvements of the surrogate over the exact prior. We also demonstrate how our proposed approach achieves higher-quality image reconstructions than methods that deviate from true Bayesian inference.

## 2 Related work

### 2.1 Bayesian inverse imaging

Image reconstruction can be framed as an inverse problem: a hidden image $\mathbf{x}^* \in \mathbb{R}^D$ must be recovered from measurements $\mathbf{y} \in \mathbb{R}^M$, where

$$\mathbf{y} = f(\mathbf{x}^*) + \epsilon. \tag{1}$$

It is usually assumed that the forward model $f : \mathbb{R}^D \to \mathbb{R}^M$ is known and that the measurement noise $\epsilon \in \mathbb{R}^M$ is a random variable with a known distribution. With an ill-posed inverse problem, there is inherent uncertainty in image reconstruction.

Bayesian imaging accounts for the uncertainty by formulating a posterior distribution $p(\mathbf{x} \mid \mathbf{y})$. The posterior can be decomposed into a likelihood term and a prior term:

$$\log p(\mathbf{x} \mid \mathbf{y}) = \log p(\mathbf{y} \mid \mathbf{x}) + \log p(\mathbf{x}) + \text{const.} \tag{2}$$

Given a log-likelihood function $\log p(\mathbf{y} \mid \mathbf{x})$ and a prior log-probability function $\log p(\mathbf{x})$, we can use established techniques for sampling from the posterior, such as Markov chain Monte Carlo (MCMC) [3] or variational inference [2]. MCMC algorithms generate a Markov chain whose stationary distribution is the posterior, but they are generally slow to converge for high-dimensional data like images. Variational inference instead approximates the posterior with a tractable distribution (e.g., Gaussian). The variational distribution is usually parameterized and thus can be efficiently optimized to represent high-dimensional data distributions. Deep Probabilistic Imaging (DPI) [25; 26] proposed an efficient variational-inference approach specifically for computationtal imaging with traditional regularizers; in DPI, the variational distribution is a discrete normalizing flow [15], which is an invertible generative model capable of representing complex distributions.

## 2.2 Diffusion models for inverse problems

Primarily developed for image generation, diffusion models [18; 12; 20; 21; 23] learn to model a rich image distribution that could be useful as a prior for image reconstruction. A diffusion model generates an image by starting from an image of noise and gradually denoising it until it becomes a clean image. We discuss this process, known as *reverse diffusion*, in more detail in Sec. 3.1.

Given an inverse problem, simply adapting a pretrained diffusion model to sample from the posterior instead of the learned prior is intractable [10]. Therefore, most diffusion-based approaches do not infer a true Bayesian posterior. Some methods project images onto a measurement-consistent subspace [24; 8; 6; 5; 7], but the projection does not account for measurement noise and might pull images away from a true posterior. Other methods follow a gradient toward higher likelihood throughout reverse diffusion [9; 13; 11; 14; 1; 19; 17], but these methods heavily approximate the posterior. Overall, these diffusion-based methods require hyperparameter-tuning to balance the measurements and the prior. As soon as hyperparameters are introduced, there is no guarantee of sampling from a posterior that represents the true uncertainty.

**Score-based priors.** Alternatively, a score-based diffusion model can be turned into a standalone, probabilistic prior (*score-based prior*) that can be paired with any measurement-likelihood function and plugged into established Bayesian-inference approaches. Feng et al. [10] proposed to do this with a log-density function based on the ODE associated with reverse diffusion (see Sec. 3.2). This function provides the log-probability of any image under the diffusion model's generative distribution, but it is computationally expensive to evaluate. When used in iterative optimization algorithms, it incurs prohibitively high time and memory costs.

# 3 Background

In this section, we review background on score-based diffusion models with an emphasis on evaluating probabilities of images with a pretrained diffusion model. We then describe how a diffusion process gives rise to an efficient denoising-based lower-bound on these image probabilities.

## 3.1 Score-based diffusion models

The core idea of a diffusion model is that it transforms a simple distribution $\pi$ to a complex image distribution through a gradual process. In this work, we follow the popular framework of denoising diffusion models, which transform noise samples from $\pi = \mathcal{N}(\mathbf{0}, \mathbf{I})$ to clean samples from the data distribution $p_{\text{data}}$ through gradual denoising. With knowledge of the noise distribution and the denoising process, we can assess the probability of a novel image under this generative model.

The transformation from a simple distribution to a complex one occurs over many steps. To determine how the data distribution should look at each step of the denoising process, we turn to a stochastic differential equation (SDE) that describes a diffusion process from clean images to noise. The diffusion SDE is defined on the time interval $t \in [0, T]$ and has the form

$$d\mathbf{x} = \mathbf{f}(\mathbf{x}, t) + g(t)d\mathbf{w}, \tag{3}$$

where $\mathbf{w} \in \mathbb{R}^D$ denotes Brownian motion. $g(t) \in \mathbb{R}$ is the diffusion coefficient, which controls the rate of noise increase. $\mathbf{f}(\cdot, t) : \mathbb{R}^D \to \mathbb{R}^D$ is the drift coefficient, which controls the deterministic evolution of $\mathbf{x}(t)$. By defining a stochastic trajectory $\{\mathbf{x}(t)\}_{t \in [0, T]}$, this SDE gives rise to a time-dependent probability distribution $p_t$, which is the marginal distribution of $\mathbf{x}(t)$. We construct $\mathbf{f}(\cdot, t)$ and $g(t)$ so that if $p_0 = p_{\text{data}}$, then $p_T \approx \pi$. Image generation amounts to reversing the diffusion, which requires the gradient of the data log-density (*score*) at every noise level in order to nudge images toward high probability under $p_{\text{data}}$. A convolutional neural network $\mathbf{s}_\theta$ known as a *score model* is trained to approximate the true score: $\mathbf{s}_\theta(\mathbf{x}, t) \approx \nabla_{\mathbf{x}} \log p_t(\mathbf{x})$.

## 3.2 Image probabilities under a score-based diffusion model

Once trained, $\mathbf{s}_\theta(\mathbf{x}, t)$ is used in a reverse-diffusion process to generate clean images from noise. The generated image distribution theoretically assigns a probability density to every possible image. However, reverse diffusion does not lead to an image distribution with tractable probabilities. In this

109 subsection, we describe two workarounds: one based on an ordinary differential equation (ODE) and
110 the other based on a denoising score-matching objective.

111 **Sampling with a reverse-time SDE.** Reversing diffusion (Eq. 3) with a score model $\mathbf{s}_\theta(\mathbf{x}, t)$ results
112 in a distribution $p_\theta^{\text{SDE}}$, denoted as such because it is determined by a reverse-time SDE:

$$\mathrm{d}\mathbf{x} = \left[\mathbf{f}(\mathbf{x}, t) - g(t)^2 \mathbf{s}_\theta(\mathbf{x}, t)\right] \mathrm{d}t + g(t)\mathrm{d}\bar{\mathbf{w}}. \tag{4}$$

113 $\bar{\mathbf{w}} \in \mathbb{R}^D$ denotes Brownian motion, and $\mathbf{f}(\cdot, t)$ and $g(t)$ are the same as in Eq. 3. To generate an
114 image, we first sample $\mathbf{x}(T) \sim \mathcal{N}(\mathbf{0}, \mathbf{I})$ and then numerically solve the reverse-time SDE for $\mathbf{x}(0)$.
115 $p_\theta^{\text{SDE}}$ is the marginal distribution of $\mathbf{x}(0)$, which for a well-trained score model is close to $p_{\text{data}}$.

116 To compute the probability of an image $\mathbf{x}$ under $p_\theta^{\text{SDE}}$, we need to invert this image from $\mathbf{x}(0) = \mathbf{x}$ to
117 $\mathbf{x}(T)$. However, this is not tractable through the SDE: just as it is intractable to reverse a random
118 walk, it is intractable to account for all the possible starting points $\mathbf{x}(T)$ that could have resulted in
119 $\mathbf{x}(0)$ through the stochastic process. Probability computation calls for an invertible process that lets
120 us map any point from $p_{\text{data}}$ to $\mathcal{N}(\mathbf{0}, \mathbf{I})$ and vice versa.

121 **Computing probabilities with an ODE.** The *probability flow ODE* [23] defines an invertible
122 sampling function for an image distribution $p_\theta^{\text{ODE}}$ theoretically the same as $p_\theta^{\text{SDE}}$. It is given by

$$\frac{\mathrm{d}\mathbf{x}}{\mathrm{d}t} = \mathbf{f}(\mathbf{x}, t) - \frac{1}{2}g(t)^2 \mathbf{s}_\theta(\mathbf{x}, t) =: \tilde{\mathbf{f}}_\theta(\mathbf{x}, t). \tag{5}$$

123 The absence of Brownian motion makes it possible to solve this ODE in both directions of time. To
124 compute the log-probability of an image $\mathbf{x}$, we map $\mathbf{x}(0) = \mathbf{x}$ to its corresponding noise image $\mathbf{x}(T)$.
125 Under the framework of neural ODEs [4], the log-probability is given by the log-probability of $\mathbf{x}(T)$
126 under $\mathcal{N}(\mathbf{0}, \mathbf{I})$ plus a normalization factor accounting for the change in density through time:

$$\log p_\theta^{\text{ODE}}(\mathbf{x}(0)) = \log \pi(\mathbf{x}(T)) + \int_0^T \nabla \cdot \tilde{\mathbf{f}}_\theta(\mathbf{x}(t), t)\mathrm{d}t, \quad \mathbf{x}(0) = \mathbf{x}, \tag{6}$$

127 Although tractable to evaluate with an ODE solver, this log-probability function is computationally
128 expensive, requiring hundreds to thousands of discrete ODE time steps to accurately evaluate.
129 Additional time and memory costs are incurred by backpropagation through the ODE and Hutchinson-
130 Skilling trace estimation of the divergence.

131 **Equivalence of $p_\theta^{\text{SDE}}$ and $p_\theta^{\text{ODE}}$.** Song et al. [22] proved that if $\mathbf{s}_\theta(\mathbf{x}, t) \equiv \nabla_\mathbf{x} \log p_t(\mathbf{x}, t)$ for all $t \in$
132 $[0, T]$ and $p_T = \pi$, then $p_\theta^{\text{ODE}} = p_\theta^{\text{SDE}} = p_{\text{data}}$. In our work, we assume that $\mathbf{s}_\theta(\mathbf{x}, t) \approx \nabla_\mathbf{x} \log p_t(\mathbf{x}, t)$
133 for almost all $\mathbf{x} \in \mathbb{R}^D$ and $t \in [0, T]$ and that $p_T \approx \mathcal{N}(\mathbf{0}, \mathbf{I})$, so that $p_\theta^{\text{ODE}} \approx p_\theta^{\text{SDE}} \approx p_{\text{data}}$. This
134 assumption empirically performed well in previous work that appealed to $p_\theta^{\text{ODE}}$ as the exact probability
135 distribution of the diffusion model [10; 23].

### 3.3 Evidence lower bound of a score-based diffusion model

137 In lieu of an exact log-probability function, Song et al. [22] derived an evidence lower-bound for
138 $p_\theta^{\text{SDE}}$ such that $b_\theta^{\text{SDE}}(\mathbf{x}) \leq \log p_\theta^{\text{SDE}}(\mathbf{x})$ for any proposed image $\mathbf{x}$. Essentially, this lower-bound
139 corresponds to how well the diffusion model is able to denoise a given image: an image with high
140 probability under the diffusion model is easy to denoise, whereas a low-probability image is difficult.

141 The lower-bound, or the negative "denoising score-matching loss" [22], is defined as

$$b_\theta^{\text{SDE}}(\mathbf{x}) := \mathbb{E}_{p_{0T}(\mathbf{x}'|\mathbf{x})}\left[\log \pi(\mathbf{x}')\right] - \frac{1}{2}\int_0^T g(t)^2 h(t)\mathrm{d}t, \tag{7}$$

142 where

$$h(t) := \mathbb{E}_{p_{0t}(\mathbf{x}'|\mathbf{x})}\left[\left\|\mathbf{s}_\theta(\mathbf{x}', t) - \nabla_{\mathbf{x}'} \log p_{0t}(\mathbf{x}' \mid \mathbf{x})\right\|_2^2 - \left\|\nabla_{\mathbf{x}'} \log p_{0t}(\mathbf{x}' \mid \mathbf{x})\right\|_2^2 - \frac{2}{g(t)^2}\nabla_{\mathbf{x}'} \cdot \mathbf{f}(\mathbf{x}', t)\right]. \tag{8}$$

143 $p_{0t}(\mathbf{x}' \mid \mathbf{x})$ denotes the transition distribution from $\mathbf{x}(0) = \mathbf{x}$ to $\mathbf{x}(t) = \mathbf{x}'$. For a drift coefficient
144 that is linear in $\mathbf{x}$, this transition distribution is Gaussian: $p_{0t}(\mathbf{x}' \mid \mathbf{x}) = \mathcal{N}(\mathbf{x}'; \alpha(t)\mathbf{x}, \beta(t)^2\mathbf{I})$. This
145 means that the gradient $\nabla_{\mathbf{x}'} \log p_{0t}(\mathbf{x}' \mid \mathbf{x})$ is directly proportional to the Gaussian noise that is
146 subtracted from $\mathbf{x}'$ to get $\mathbf{x}$. Eq. 7 is efficient to compute since we can evaluate it by adding Gaussian

147 noise to $\mathbf{x}$ without having to solve an initial-value problem as with the ODE. In fact, Eq. 7 is closely
148 related to the denoising score-matching objective used to efficiently train diffusion models [23].

149 Intuitively, we can interpret Eq. 7 as associating an image's probability with how well the score model
150 $\mathbf{s}_\theta$ could denoise that image if it underwent diffusion. This is represented by the first term in $h(t)$
151 (Eq. 8). To assess the probability of an image $\mathbf{x}$, we perturb it with Gaussian noise to get $\mathbf{x}'$ and then
152 ask the score model to estimate the noise that was added. If $\mathbf{s}_\theta(\mathbf{x}, t)$ accurately estimates the noise,
153 then $\|\mathbf{s}_\theta(\mathbf{x}', t) - \nabla_{\mathbf{x}'} \log p_{0t}(\mathbf{x}' \mid \mathbf{x})\|_2^2$ is small, and the value of $b_\theta^{\mathrm{SDE}}(\mathbf{x})$ becomes larger.

154 The remaining terms in $h(t)$ are normalizing factors independent of $\theta$. The term $\mathbb{E}_{p_{0T}(\mathbf{x}'|\mathbf{x})}\left[\log \pi(\mathbf{x}')\right]$
155 accounts for the probabilities of the noise images $\mathbf{x}(T)$ that could result from $\mathbf{x}$ being entirely diffused.

## 4 Method

157 Inspired by previous theoretical work [22], we propose $b_\theta^{\mathrm{SDE}}$ as an efficient surrogate prior for the
158 exact score-based prior in Bayesian imaging. In this section, we describe our approach for efficient
159 posterior inference with a score-based prior.

### 4.1 Variational inference with a surrogate score-based prior

161 Given measurements $\mathbf{y} \in \mathbb{R}^M$ (with a known log-likelihood function) and a score-based diffusion
162 model (parameterized by $\theta$) as the prior, our goal is to sample from the image posterior $p_\theta(\mathbf{x} \mid \mathbf{y})$.
163 We follow a variational-inference approach by optimizing the parameters of a variational distribution
164 to closely approximate the target posterior.

165 Let $q_\phi$ denote the variational distribution with parameters $\phi$, and we assume $q_\phi$ to have tractable
166 log-probabilities. We optimize $\phi$ to minimize the KL divergence from $q_\phi$ to the target posterior:

$$\phi^* = \arg\min_\phi D_{\mathrm{KL}}(q_\phi \| p_\theta(\cdot \mid \mathbf{y})) = \arg\min_\phi \mathbb{E}_{\mathbf{x} \sim q_\phi}\left[-\log p(\mathbf{y} \mid \mathbf{x}) - \log p_\theta^{\mathrm{ODE}}(\mathbf{x}) + \log q_\phi(\mathbf{x})\right]. \quad (9)$$

167 $q_\phi$ can be various types of distributions. For example, it could be a Gaussian distribution with a
168 diagonal covariance matrix so that $\phi := [\mu^\top, \sigma^\top]^\top$, where $\mu \in \mathbb{R}^D$ and $\sigma \in \mathbb{R}^D$ ($\sigma > \mathbf{0}$) are
169 the mean and pixel-wise standard deviation. As DPI showed [25], $q_\phi$ could also be a RealNVP
170 normalizing flow with network parameters $\phi$.

171 To circumvent the computational challenges of evaluating the prior term $\log p_\theta^{\mathrm{ODE}}(\mathbf{x})$, we replace it
172 with the surrogate $b_\theta^{\mathrm{SDE}}(\mathbf{x})$. This results in the following objective:

$$\phi^* = \arg\min_\phi \mathbb{E}_{\mathbf{x} \sim q_\phi}\left[-\log p(\mathbf{y} \mid \mathbf{x}) - b_\theta^{\mathrm{SDE}}(\mathbf{x}) + \log q_\phi(\mathbf{x})\right]. \quad (10)$$

173 We can also think of $b_\theta^{\mathrm{SDE}}$ as replacing the intractable $\log p_\theta^{\mathrm{SDE}}$ in Eq. 9. Since $-\log p_\theta^{\mathrm{SDE}} \leq -b_\theta^{\mathrm{SDE}}$,
174 our surrogate objective minimizes the upper-bound of a valid KL divergence involving $p_\theta^{\mathrm{SDE}}$.

### 4.2 Implementation details

176 **Evaluating $b_\theta^{\mathbf{SDE}}(\mathbf{x})$.** The formula for $b_\theta^{\mathrm{SDE}}(\mathbf{x})$ (Eq. 7) contains a time integral and expectation
177 over $p_{0t}(\mathbf{x}' \mid \mathbf{x})$ that can be estimated with numerical methods. Following Song et al. [22], we use
178 importance sampling with time samples $t \sim p(t)$ for the time integral and Monte-Carlo approximation
179 with noisy images $\mathbf{x}' \sim \mathcal{N}(\alpha(t)\mathbf{x}, \beta(t)^2\mathbf{I})$ for the expectation. The proposal distribution $p(t) :=$
180 $\frac{g(t)^2}{\beta(t)^2 Z}$ was empirically verified to result in lower variance in the estimation of $b_\theta^{\mathrm{SDE}}(\mathbf{x})$ [22]. We
181 provide the following formula used in our implementation, which estimates the time integral with
182 importance sampling and the expectation with Monte-Carlo approximation, for reference:

$$b_\theta^{\mathrm{SDE}}(\mathbf{x}) \approx \frac{1}{N_z} \sum_{j=1}^{N_z} \log \pi(\mathbf{x}'_j)$$

$$- \frac{1}{2N_t N_z} \sum_{i=1}^{N_t} Z\beta(t)^2 \sum_{j=1}^{N_z} \left[\left\|\mathbf{s}_\theta(\mathbf{x}'_{ij}, t_i) + \frac{\mathbf{z}_{ij}}{\beta(t_i)}\right\|_2^2 - \left\|\frac{\mathbf{z}_{ij}}{\beta(t_i)}\right\|_2^2 - \frac{2}{g(t_i)^2}\nabla_{\mathbf{x}'_{ij}} \cdot \mathbf{f}(\mathbf{x}'_{ij}, t_i)\right]$$

$$\text{s.t.} \quad t_i \sim p(t), \ \mathbf{z}_{ij} \sim \mathcal{N}(\mathbf{0}, \mathbf{I}), \ \mathbf{x}'_{ij} = \alpha(t_i)\mathbf{x} + \beta(t_i)\mathbf{z}_{ij}, \ \mathbf{x}'_j \sim \mathcal{N}(\alpha(T)\mathbf{x}, \beta(T)^2\mathbf{I})$$

$$\forall\, i = 1, \ldots, N_t, j = 1, \ldots, N_z. \quad (11)$$

183 $N_t$ is the number of time samples used to approximate the time integral, and $N_z$ is the number
184 of noise samples taken to approximate the expectation over $p_{0t}(\mathbf{x}' \mid \mathbf{x})$. In our experiments, we
185 set $N_t = N_z = 1$. Increasing the number of time and noise samples does not efficiently decrease
186 variance in the estimated value of $b_\theta^{\text{SDE}}(\mathbf{x})$. We use the Variance Preserving (VP) SDE.

187 **Optimization.** We use stochastic gradient descent to optimize $\phi$, Monte-Carlo approximating the
188 expectation in Eq. 10 with a batch of $\mathbf{x} \sim q_\phi$. We find that estimating $b_\theta^{\text{SDE}}(\mathbf{x})$ has higher variance
189 than estimating $\log p_\theta^{\text{ODE}}(\mathbf{x})$. For example, in Fig. 4, $b_\theta^{\text{SDE}}(\mathbf{x})$ with $N_t = 2048, N_z = 1$ shows higher
190 variance than $\log p_\theta^{\text{ODE}}(\mathbf{x})$ with 16 trace estimators. When optimizing a complex distribution like
191 RealNVP, a lower learning-rate helps mitigate training instabilities caused by variance. For example,
192 in Fig. 3b the learning rate with the exact prior was 0.0002, while the learning rate with the surrogate
193 prior was 0.00001. Please refer to the supplemental text for more optimization details.

# 5 Experiments

195 We validate our proposed approach on the tasks of accelerated MRI, image denoising, and reconstruc-
196 tion from low spatial frequencies. We highlight accelerated (or compressed sensing) MRI because in
197 addition to being a real-world imaging problem that calls for accurate posterior estimation, it is the
198 focus of much related work [24; 13]. In MRI, measurements in a spatial-frequency space ($\kappa$-space) are
199 obtained to help reveal a hidden anatomical image. Accelerated MRI reduces the number of $\kappa$-space
200 measurements, thus reducing the scan time but also making the image reconstruction ill-posed. The
201 supplemental text provides details on how measurements were generated for all tasks.

## 5.1 Efficiency improvements

203 In Tab. 1 and Fig. 2, we quantify the efficiency improvements
204 of the surrogate prior for an accelerated MRI task at different
205 image resolutions. We drew a test image from the fastMRI knee
206 dataset [27] and resized it to $16 \times 16$, $32 \times 32$, $64 \times 64$, $128 \times$
207 $128$, and $256 \times 256$. For each image size, we trained a score
208 model on training images of the corresponding size from the
209 fastMRI dataset of single-coil knee scans. We then optimized a
210 Gaussian distribution with diagonal covariance to approximate
211 the posterior. The batch size was 64 for the surrogate and 32 for
212 the exact prior (a smaller batch size was needed to fit $64 \times 64$
213 optimization into GPU memory). Convergence was defined
214 by setting a minimum acceptable change in the mean of the
215 estimated posterior between optimization steps.

| Image size | Surrogate | Exact |
|---|---|---|
| $16 \times 16$ | 0.029 | 19.5 |
| $32 \times 32$ | 0.038 | 41.9 |
| $64 \times 64$ | 0.090 | 123 |
| $128 \times 128$ | 0.294 | N/A |
| $256 \times 256$ | 1.115 | N/A |

Table 1: Iteration time [sec/step]. Each iteration of gradient-based optimization of the variational distribution is 2 to 3 orders of magnitude faster with the surrogate prior.

216 We find at least two orders of magnitude in time improvement
217 with the surrogate prior. Tab. 1 compares the iteration time
218 between the two priors. Fig. 2 compares the total time it takes to optimize the variational distribution.
219 The surrogate also significantly improves memory consumption, which in turn enables optimizing
220 higher-dimensional posteriors. Following standard practice, we just-in-time (JIT) compile the
221 optimization step to reduce time/step at the cost of GPU memory. Fig. 2 shows how the surrogate
222 prior significantly reduces memory requirements and scales better with image size. The exact prior
223 could only handle up to $32 \times 32$ before exceeding GPU memory (we tested on 4x 48GB GPUs).
224 While memory could be reduced with a smaller batch size, this would make optimization more time-
225 consuming. On the other hand, our surrogate prior supports much larger images, as we demonstrate in
226 Fig. 1 for $256 \times 256$[1] MRI with a Gaussian-approximated posterior. This type of principled inference
227 of high-dimensional image posteriors was not possible before with the exact score-based prior.

## 5.2 Posterior estimation under the surrogate vs. exact prior

229 We cannot expect the surrogate prior $b_\theta^{\text{SDE}}$ to be an identical substitute for the exact prior $\log p_\theta^{\text{ODE}}$.
230 Importantly, though, we verify in Fig. 3a that both the surrogate and the exact prior recover a ground-
231 truth Gaussian posterior derived from a Gaussian likelihood and prior. The variational distribution

---

[1]Larger images may be feasible but are more memory-intensive, which imposes more restrictions on the batch size and the complexity of the variational distribution.

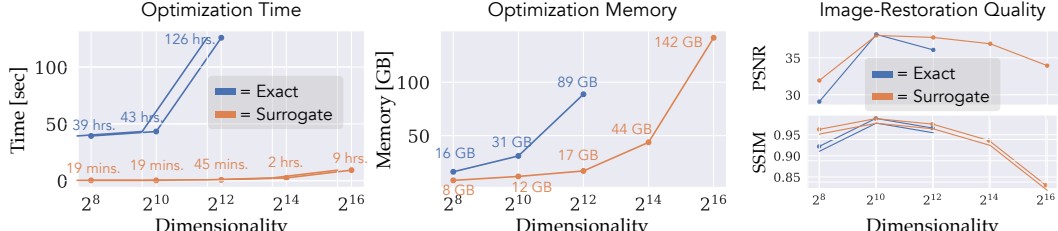

Figure 2: Computational efficiency of our proposed surrogate prior ("Surrogate") vs. exact prior ("Exact"). For each image size, we estimated a posterior of images conditioned on $4\times$-accelerated MRI measurements of a knee image, using a Gaussian distribution with diagonal covariance as the variational distribution. The hardware is 4x NVIDIA RTX A6000. The surrogate prior allows for variational inference of image sizes that are prohibitively large for the exact prior. For image sizes supported by the exact prior, the surrogate improved total optimization time by over $120\times$ while using less memory and scaling better with image size. "Image-Restoration Quality" verifies that optimization with the surrogate was done fairly, as the PSNR and SSIM of the converged posterior (averaged over 128 samples) are at least as high as with the exact prior.

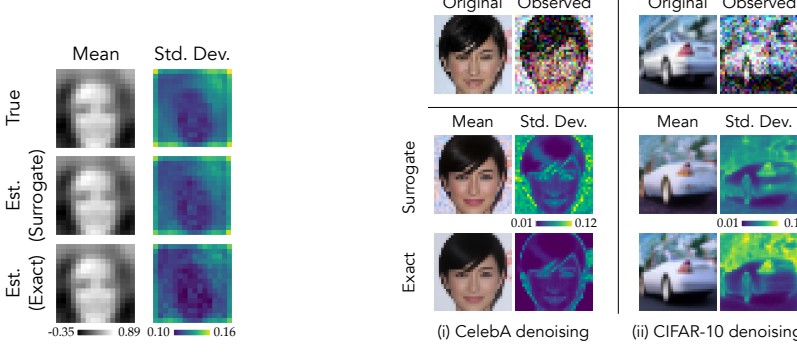

(a) Ground-truth (Gaussian) posterior.

(b) Complex posteriors.

Figure 3: Estimated posteriors under surrogate vs. exact prior. For each task, the variational distribution is a RealNVP, and the score model is the same between both prior functions. **(a)** Both prior functions recover the correct (Gaussian) posterior. The score-based prior was trained on samples from a known Gaussian distribution (originally fit to $16 \times 16$ face images), and the measurements are the lowest 6.25% spatial frequencies of a test image from the prior. Since the prior and likelihood are both Gaussian, we know the ground-truth Gaussian posterior. **(b)** We estimate posteriors for (i) denoising a CelebA image and (ii) denoising a CIFAR-10 image. The score-based prior was trained on CelebA in (i) and CIFAR-10 in (ii). Visual differences between the estimated posteriors appear mostly in the image background, and the prior functions result in comparable image quality.

used for inference is a RealNVP, and the score model (used by both the surrogate and exact prior) was trained on samples from the known Gaussian prior.

Nonetheless, the surrogate could result in a different locally-optimal variational posterior, particularly if the posterior is complex with various local minima in the variational objective. Fig. 3b compares posteriors (with unknown true distribution) approximated by a RealNVP under the surrogate versus exact prior. For each task (CelebA denoising and CIFAR-10 denoising), both prior functions used the same pretrained score model. We observe in these comparisons that most of the differences appear in the image background and that both priors result in a plausible mean reconstruction and uncertainty.

**Visualizing the bound bap throughout optimization** helps shed light on why the two priors converge to different solutions even if the underlying score model is the same. Fig. 4 shows probabilities of samples generated by $q_\phi$ (in this case, a RealNVP) as optimization progresses. At each checkpoint of $q_\phi$, we plot $\log p_\theta^{\text{ODE}}(\mathbf{x})$ versus $b_\theta^{\text{SDE}}(\mathbf{x})$ (approximated with $N_t = 2048$ for reduced variance) for samples $\mathbf{x} \sim q_\phi$ coming from both the exact and surrogate optimization of $q_\phi$. Importantly, we find that the surrogate is a valid bound for the ODE log-density: $b_\theta^{\text{SDE}}(\mathbf{x}) \leq \log p_\theta^{\text{ODE}}(\mathbf{x})$ for all

$\mathbf{x} \sim q_\phi(\mathbf{x})$, except for some outliers due to variance of $b_\theta^{\text{SDE}}(\mathbf{x})$. However, we find that optimization follows a different trajectory depending on the prior. With the surrogate, samples $\mathbf{x} \sim q_\phi$ tend toward a region where the bound gap is small (i.e., $b_\theta^{\text{SDE}}(\mathbf{x})$ is close to $\log p_\theta^{\text{ODE}}(\mathbf{x})$). Meanwhile, the exact prior follows a loss landscape whose structure appears to be independent of the lower-bound. Note that samples from $q_\phi$ optimized under the exact prior obtain higher values of $b_\theta^{\text{SDE}}(\mathbf{x})$ than samples obtained under the surrogate. The observations in Fig. 4 suggest that gradients under the surrogate tend to push the $q_\phi$ distribution along the boundary of equality between $b_\theta^{\text{SDE}}$ and $\log p_\theta^{\text{ODE}}$. This constrains the path taken through gradient descent and subsequently the converged solution.

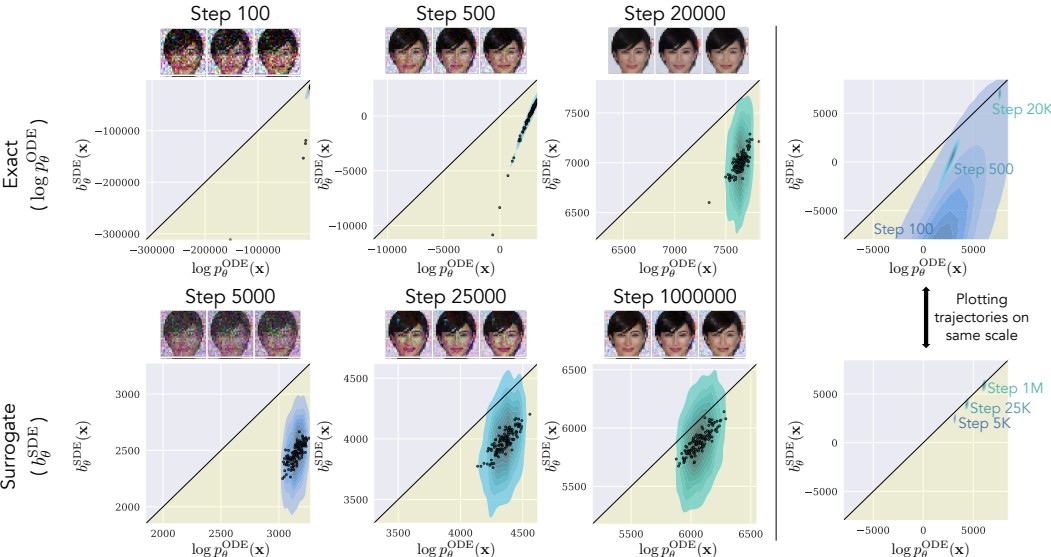

Figure 4: $b_\theta^{\text{SDE}}(\mathbf{x})$ vs. $\log p_\theta^{\text{ODE}}(\mathbf{x})$ for samples $\mathbf{x} \sim q_\phi$ as optimization of $\phi$ progresses. The task is from Fig. 3b(i). For each plot, we took 128 samples $\mathbf{x} \sim q_\phi$ and performed 20 estimates each of $b_\theta^{\text{SDE}}(\mathbf{x})$ and $\log p_\theta^{\text{ODE}}(\mathbf{x})$. The density map is a KDE plot of all $128 \cdot 20 = 2560$ values; the 128 scatter points represent the mean estimate for each $\mathbf{x}$. The black line indicates perfect agreement between $b_\theta^{\text{SDE}}(\mathbf{x})$ and $\log p_\theta^{\text{ODE}}(\mathbf{x})$. We expect all points to lie below this black line for $b_\theta^{\text{SDE}}$ to be a lower-bound. We find that $b_\theta^{\text{SDE}}(\mathbf{x}) \leq \log p_\theta^{\text{ODE}}(\mathbf{x})$ (up to variance error), but the optimization progresses differently depending on the prior. Gradients under the surrogate push $q_\phi(\mathbf{x})$ along the black line to increase $b_\theta^{\text{SDE}}(\mathbf{x})$ without exceeding $\log p_\theta^{\text{ODE}}(\mathbf{x})$. Optimization under the exact prior proceeds more freely, although eventually achieves higher $b_\theta^{\text{SDE}}(\mathbf{x})$ at convergence. This visualization may help explain differences in the posterior estimated with the surrogate vs. exact prior.

## 5.3 Image-reconstruction quality

It would be reasonable to assume that diffusion-based approaches discussed in Sec. 2, although less principled, may lead to better visual quality than a Bayesian approach. However, we find that in addition to providing more-reliable uncertainty, our approach achieves higher-fidelity reconstructions. We note that similarity to a ground-truth image does not indicate a correct posterior. Still, for a good prior, it might be desirable for posterior samples to accurately reflect the true underlying image.

We performed multiple MRI tasks at different acceleration rates and compared our approach to three baselines: **SDE+Proj** [24], **Score-ALD** [13], and Diffusion Posterior Sampling (**DPS**) [9]. SDE+Proj projects images onto a measurement subspace. Score-ALD and DPS approximate the posterior throughout reverse diffusion. All baselines involve at least one measurement-weight hyperparameter. The implementations and hyperparameter settings for SDE+Proj and Score-ALD were provided by Song et al. [24]. For DPS, we followed the implementation of Chung et al. [9] and performed a hyperparameter search on an $8\times$-acceleration test image to find the optimal PSNR.

We simulated MRI at three different acceleration factors for ten test images, resulting in thirty posterior distributions to be estimated. As baseline implementations do not account for measurement noise, we gave the baselines noiseless measurements and set a near-zero measurement noise for our method. The test images were randomly sampled from the fastMRI dataset and resized to $64 \times 64$.

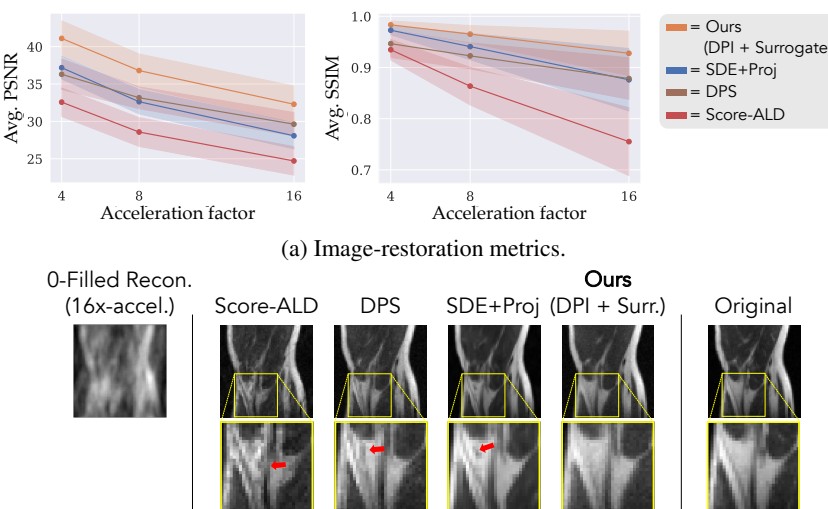

(a) Image-restoration metrics.

(b) Example image reconstructions for $16\times$ acceleration.

Figure 5: Accelerated MRI of knee images. **(a)** For each acceleration factor ($4\times$, $8\times$, $16\times$), we estimated posteriors for ten images measured at that acceleration rate. Baseline methods do not capture a true posterior: Score-ALD and DPS strongly approximate the posterior uncertainty, and SDE+Proj is a non-Bayesian projection-based approach. For each method, we computed the average PSNR and SSIM of 128 estimated posterior samples. The line plot shows the average result across the ten tasks; the shaded region shows one std. dev. above and below the average. **(b)** An example of $16\times$-accel. MRI. The cropped region exemplifies how baselines hallucinate incorrect more features than necessary. (a) and (b) are evidence that a principled Bayesian approach can capture a more accurate posterior than previous unsupervised methods.

Our approach was DPI with the surrogate prior, meaning we optimized a RealNVP to approximate each posterior and used the lower-bound function $b_\theta^{\text{SDE}}$ as the prior log-density. The score model $\mathbf{s}_\theta$ was trained on $64 \times 64$ images of knee scans from fastMRI and stayed fixed across all methods.

Our method achieves a marked improvement in PSNR and SSIM over the three baselines (Fig. 5). Across all acceleration factors and baselines, our method improves PSNR by between $2.7$ and $8.5$ dB. Even though each method uses the same score model, restoration quality depends on how the prior is used for inference; whereas baselines loosely approximate the posterior and involve hyperparameters, our approach treats the diffusion model as a standalone prior in Bayesian inference.

# 6 Conclusion

We have presented a surrogate function that provides efficient access to score-based priors for Bayesian inference. We empirically verify that the evidence lower-bound $b_\theta^{\text{SDE}}(\mathbf{x}) \leq \log p_\theta^{\text{SDE}}(\mathbf{x})$ can serve as a proxy for evaluating the log-prior of an image under a trained diffusion model. Paired with any log-likelihood function, $b_\theta^{\text{SDE}}(\mathbf{x})$ can be plugged into a Bayesian-inference algorithm. Our experiments with variational inference show at least two orders of magnitude in runtime improvement and significant memory improvement over the ODE-based prior. This enables inference of images previously too large for a strictly Bayesian approach, such as $256 \times 256$ pixels. We also establish that a principled approach like ours outperforms baselines on image-restoration metrics, evidence that following a Bayesian approach results in more-reliable image reconstructions.

**Limitations.** A variational approach like ours depends on the expressiveness of the variational distribution. Improvements may be possible by using a diffusion model instead of a discrete normalizing flow as the variational distribution. We also note that there are open theoretical questions about $b_\theta^{\text{SDE}}$ as it relates to $p_\theta^{\text{ODE}}$ [16]. **Broader impact.** Our proposed framework for efficient estimation of high-dimensional, sophisticated posteriors has broad potential impact for computational imaging. Many imaging tasks, especially in science and medicine, would benefit from accurate uncertainty quantification with principled, data-driven priors.

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
