# OpenReview forum: "Efficient Bayesian Computational Imaging with a Surrogate Score-Based Prior"
_NeurIPS.cc/2023/Conference — Submitted to NeurIPS 2023_

### Official Review · Reviewer_sVNv · 2023-07-06

**Soundness:** 3 good
**Presentation:** 2 fair
**Contribution:** 2 fair
**Rating:** 4
**Confidence:** 4

**Summary:**

The paper deals with variational inference (VI) of the posterior distribution when using a diffusion based generative model as the prior distribution. The paper proposes using a lower bound on the log probability of the prior distribution (instead of calculating it through the  ODE in standard diffusion based generative models as in [1]) for the optimization of the Kullback-Leibler (KL) divergence between in the VI framework. This results in a more efficient training procedure for the VI than using the true log probability as well as a more efficient memory footprint.

The authors then validate their method using an MRI dataset as well as the most common CelebA and Cifar datasets.

[1] Berthy T Feng, Jamie Smith, Michael Rubinstein, Huiwen Chang, Katherine L Bouman, and William T318
Freeman. Score-based diffusion models as principled priors for inverse imaging. arXiv preprint319
arXiv:2304.11751, 2023

**Strengths:**

* The paper proposes a solution to estimating the KL that is indeed way faster than the previous known methods and this allows for **effective gains in training time and memory footprints**. They provide some visual evidence that those gains do not alter considerably the quality of the variational approximation of the posterior distribution.

* The presentation of the problem is good and they motivate the need for an efficient algorithm with a real word application (Accelerated MRI).

* The explanation of the method is clear and reproducible. I expect practitioners would be able to clearly implement the algorithm with the description given in the main part of the text.

**Weaknesses:**

* The paper for me gives a slightly overstated presentation of the variational inference as a sampling from a true Bayesian posterior. The Bayesian posterior is clearly defined once one states that the prior is the diffusion based generative model. Therefore, when doing Variational Inference, we are only approximating (to an unknown degree) this posterior distribution, so I no way the samples generated by the outcome distribution of the VI procedure are not an approximation of the Bayesian problem.  Of course, VI is still a useful approach, but I would say that I agree that the proposed algorithm is closer to the "true Bayesian inference" (line 42) than other such as DPS [2] or SMC-DIFF [3].

* The numerical evaluation and comparison with other algorithms is insufficient.
 1) **Evaluating the distance to the true posterior**: The numeric compare only visually the posterior with the posterior obtained from [1]. When comparing with [2], the paper compare SSIM and PSNR as well as visual assessment of the reconstructions. As stated in line 258 of the paper, for ill posed inverse problems comparison to the "true image" can not be considered an adequate metric. I'd suggest considering an example where the posterior is analytically available (for example, when considering a Gaussian likelihood with a diffusion model over a mixture of Gaussians). In such case, several metrics can be used to compare the different methods ([1], [2]) such as the sliced wasserstein or even the KL.

2) **Complexity**: When comparing with other methods such as [2], we should keep in mind that the proposed method needs an optimization problem for each measurement $y$ (minimization of the KL). This is not the case for some of the "posterior diffusion samplers" such as [2] and [3]. Therefore, the actual computing time needed once we receive a measurement $y$ is smaller for [2] and [3] quite considerably.

I'd be inclined to augment my grade if the quantification of the error to the true posterior is better understood, specially in comparison with [1] and [2].



[2] Hyungjin Chung, Jeongsol Kim, Michael Thompson Mccann, Marc Louis Klasky, & Jong Chul Ye (2023). Diffusion Posterior Sampling for General Noisy Inverse Problems. In The Eleventh International Conference on Learning Representations .

[3] Brian L. Trippe, Jason Yim, Doug Tischer, David Baker, Tamara Broderick, Regina Barzilay, & Tommi S. Jaakkola (2023). Diffusion Probabilistic Modeling of Protein Backbones in 3D for the motif-scaffolding problem. In The Eleventh International Conference on Learning Representations .



**Questions:**

1. I do not understand figure 3(a). How can the prior be Gaussian and at the same time come from a diffusion generative model on CelebA? I understand that somehow the score was trained to sample from a Gaussian (if the target distribution is a Gaussian the score is analytically available), but it's not clear how. If the posterior is Gaussian, why not using a Gaussian family for VI instead of RealNVP?  Also, if everything is Gaussian, one could in principle calculate several metrics between each algorithm and the true distribution, which would be a non subjective metric for the posterior.

2. Figure 4 suggests that the minimum found by both the proposed algorithm and [1] are quite different. What is the KL gap between the two?

---

> ### Author Rebuttal · Authors · 2023-08-10
>
> Thank you for your thoughtful feedback. Please refer to the global rebuttal for discussions on computational cost and closeness to the true posterior. Below we address specific questions.
> ***
> Q: Evaluating the distance to the posterior.
>
> A: Thank you for your suggestion to quantitatively assess our method with a mixture-of-Gaussians prior. We will add such an experiment to the main or supplementary material.
> ***
> Q: Figure 3(a). How can the prior be Gaussian and at the same time come from a diffusion generative model on CelebA?
>
> A: In that experiment, the diffusion model was trained on a Gaussian approximation of CelebA. Specifically, we computed the empirical mean and covariance of CelebA training images and trained the diffusion model on samples from a Gaussian distribution with the same mean and covariance. This allows us to analytically derive the Gaussian posterior from the known Gaussian likelihood and Gaussian prior. We will clarify that experiment in the text.
> ***
> Q: Figure 4. What is the KL gap between the two?
>
> A: Let $q_{\phi_1}$ and $q_{\phi_2}$ be the variational posteriors optimized with the surrogate score-based prior and exact score-based prior, respectively, in Fig. 4. $\phi_1$ and $\phi_2$ are each the parameters of a RealNVP. The reverse KL divergence is $$\text{KL}(q_{\phi_2}\lVert q_{\phi_1})=E_{\mathbf{x}\sim q_{\phi_2}}\left[\log q_{\phi_2}(\mathbf{x})-\log q_{\phi_1}(\mathbf{x})\right]\approx 3011.2$$ (approximated using 10240 samples from the exact variational posterior). The forward KL divergence is $$\text{KL}(q_{\phi_1}\lVert q_{\phi_2})=E_{\mathbf{x}\sim q_{\phi_1}}\left[\log q_{\phi_1}(\mathbf{x})-\log q_{\phi_2}(\mathbf{x})\right]\approx 277739.5$$ (approximated using 10240 samples from the surrogate variational posterior with two outliers removed). Please note that while RealNVP normalizing flows provide tractable log-probabilities, their accuracy on out-of-distribution images is not well-proven and may affect the accuracy of KL estimates.

---

### Official Review · Reviewer_iehD · 2023-07-06

**Soundness:** 2 fair
**Presentation:** 3 good
**Contribution:** 2 fair
**Rating:** 3
**Confidence:** 4

**Summary:**

This paper focuses on solving inverse problems using diffusion based probabilistic models. The approach considered consists in minimizing the KL divergence between a variational posterior and the true posterior of the diffusion model. Computing this KL involves approximating the log probability of the diffusion model's marginal (which is assumed to approximate the true data generating distribution). A previous paper [1] used the very expensive ODE approach to approximate this log probability, making the whole method prohibitively expensive and inefficient when factoring in the computational cost. The present paper suggests instead minimizing an upper bound on the KL divergence, using a lower bound on the log probability of interest that was derived in [2].

[1] *Feng, B.T., Smith, J., Rubinstein, M., Chang, H., Bouman, K.L. and Freeman, W.T., 2023. Score-Based diffusion models as principled priors for inverse imaging. arXiv preprint arXiv:2304.11751.*

[2] *Song, Yang, Conor Durkan, Iain Murray, and Stefano Ermon. "Maximum likelihood training of score-based diffusion models." Advances in Neural Information Processing Systems 34 (2021): 1415-1428.*



**Strengths:**

The proposed method is more efficient than that proposed in [1]. It results in a drastic reduction of computational time and memory cost while maintaining having very similar performance.

**Weaknesses:**

- I believe that the KL approach is sound and reliable since the resulting variational approximation will likely not sample outside the support of the posterior. It is however extremely costly in terms of computational time and memory cost. It takes 9 hours to obtain a variational approximation over a **single image** of dimension 2^16. On the other hand, classical posterior sampling methods such as DPS [3] take less than one minute for larger images. One might then argue that such methods do not target the exact posterior and it is true! However, the approach in the present paper is also not guaranteed to sample the correct posterior; the forward KL suffers from mode collapse and posteriors over high dimensional images are highly multimodal. The only advantage of this approach is that it will likely not "hallucinate" but I do not think that it is in anyway competitive with other existing methods when one factors in the computational cost.

- The authors should have at least illustrated their method on simple toy examples where the posterior is available and multimodal, so that we can see if it indeed recovers the posterior completely.

[3] Chung, Hyungjin, Jeongsol Kim, Michael T. Mccann, Marc L. Klasky, and Jong Chul Ye. "Diffusion posterior sampling for general noisy inverse problems." ICLR (2023)

**Questions:**

- Figure 2, page 7. How come the present method scales better with image size? Shouldn't it be the opposite?

- The main reason for the huge computational cost is the fact this method requires differentiating the score network with respect to its input (by the chain rule). This is because the authors seek an approximation of the log probability. However, it seems to me that what is actually need is $\nabla_x \log p_{\theta}(x)$ (again by the chain rule). While this score is not available in practice, can't it be approximated by taking for example the score at a time $t$ close to $0$?

**Limitations:**

see above.

---

> ### Author Rebuttal · Authors · 2023-08-10
>
> Thank you for your feedback. Please refer to the global rebuttal for discussions on computational cost and closeness to the true posterior. Below we address specific questions.
> ***
> Q: Illustrate on toy examples where the posterior is available and multimodal.
>
> A: Thank you for your suggestion. We will add an experiment with a mixture-of-Gaussians posterior (in addition to the Gaussian posterior shown in Fig. 3(a)) to the main or supplementary material.
> ***
> Q: Figure 2, page 7. How come the present method scales better with image size?
>
> A: The plots show optimization time and memory (lower is better). As the plots show, computational cost increases with image size. We do find that the efficiency gap between our method and [Feng 2023] widens as the image size increases.
> ***
> Q: While this score is not available in practice, can't it be approximated by taking for example the score at a time t close to 0?
>
> A: Fig. 5 in [Feng 2023] addresses exactly this question. Approximating $\nabla_\mathbf{x}\log p_\theta(\mathbf{x})$ with the score-model output at time $t$ close to 0 leads to an incorrect posterior. One hypothesis for this behavior is that the score-model neural-network does not generalize to out-of-distribution images and is only accurate for images with similar noise levels as the ones it saw during training.
> ***
> References:
>
> [Feng 2023] B.T. Feng, J. Smith, M. Rubinstein, H. Chang, K.L. Bouman, and W.T. Freeman. Score-based diffusion models as principled priors for inverse imaging. ICCV, 2023.

---

### Official Review · Reviewer_huK6 · 2023-07-07

**Soundness:** 4 excellent
**Presentation:** 4 excellent
**Contribution:** 3 good
**Rating:** 9
**Confidence:** 5

**Summary:**

Authors propose a non-amortized variational inference approach to solve large-scale Bayesian inference problems where the prior is based on an cheap-to-evaluate approximation to a pretrained diffusion model.

**Strengths:**


**Originality.** This paper hits the nail on the head when it comes to large-scale Bayesian inference in the context of inverse problems, especially when diffusion models are used as priors.

**Quality and clarity.** The paper is well-written and easy to follow. The authors have done a great job in explaining the proposed approach and the experiments are well-designed to demonstrate the effectiveness of the approach.

**Significance.** This approach allows to leverage the power of diffusion models in approximating complex distributions in solving large-scale Bayesian inference problems. This is a significant contribution to the field and I am interested to apply this approach to another inverse problem domain.


**Weaknesses:**

* My primary concern revolves around the justification for selecting diffusion models as prior distributions over alternative generative models. It would greatly enhance the paper to thoroughly examine the advantages and disadvantages of diffusion models compared to other generative models, specifically within the framework of large-scale Bayesian inference. It would be ideal to include a comparison with amortized normalizing flows and/or injective flows. This raises the question: why not initially employ a normalizing flow to learn the the prior or full posterior distribution (amortized VI)?


**Questions:**

* I would like to know the authors' stance on amortized vs. non-amortized variational inference. What would be required to amortize their objective function, enabling the use of $q_{\phi}$ to approximate the posterior distribution for any new observation after training? This approach has the potential to make training costs offline, enabling fast posterior sampling during test time.


**Limitations:**

* It would be beneficial to provide additional comments on the limitations that arise when dealing with "out-of-distribution" data (unknown being out of diffusion model distribution).

---

> ### Author Rebuttal · Authors · 2023-08-10
>
> Thank you for your encouraging and thoughtful feedback. Below we address specific questions.
> ***
> Q: Diffusion models vs. other generative models as priors for large-scale Bayesian inference?
>
> A: We initially investigated discrete normalizing flows (NFs) and found they performed poorly as image priors. To quantify the discrepancy between an NF prior and a score-based prior, we conducted the same experiment in Fig. 3(b)(i) with an NF prior. The task was to denoise a CelebA test image (noise std. dev. = 0.2 = 20% of the image dynamic range). For the NF prior, we trained a RealNVP on the same CelebA training data that the score-based prior had been trained on. We then followed our variational-inference approach to approximate the posterior with a (separate) RealNVP. Below are the average (+/- std. dev.) PSNR and SSIM across 10240 estimated posterior samples under each prior.
>
> *Score-based diffusion model (exact log-probability as proposed in [Feng 2023]).*
>
> PSNR: 24.31 +/- 0.2
>
> SSIM: 0.88 +/- 0.01
>
> *Score-based diffusion model (our proposed surrogate).*
>
> PSNR: 22.07 +/- 0.3
>
> SSIM: 0.87 +/- 0.01
>
> *Normalizing flow.*
>
> PSNR: 11.66 +/- 0.2
>
> SSIM: 0.32 +/- 0.02
>
> These results suggest that a diffusion model is a more-effective image prior than a normalizing flow, even when using the efficient surrogate that we propose.
> ***
> Q: Non-amortized vs. amortized variational inference?
>
> A: Thank you for your insightful suggestion to investigate amortized variational inference. As you noted, amortized variational inference could adapt the estimated posterior to new measurements more efficiently. This is a promising direction for future work, especially when the imaging task involves many measurements with similar structure (such as in video reconstruction).
> ***
> Q: Out-of-distribution measurements?
>
> A: Robustness to mismatched priors is an advantage of score-based priors. In [Feng 2023], Figs. 8 and 9 demonstrate that exact score-based priors are more robust to out-of-distribution measurements than baseline methods (including [Chung&Kim 2023]). Since baselines sample with the trained diffusion model, they are heavily biased to sample from the prior. When the measurement weight is low, they hallucinate features from the prior; when the measurement weight is high, they introduce unrealistic artifacts. In contrast, score-based priors automatically find a stable posterior even when the source image is far from the prior.
>
> We have empirically found that surrogate score-based priors offer similar robustness to out-of-distribution measurements. We would be happy to add an experiment demonstrating this in our revised manuscript or supplementary material.
> ***
> References:
>
> [Feng 2023] B.T. Feng, J. Smith, M. Rubinstein, H. Chang, K.L. Bouman, and W.T. Freeman. Score-based diffusion models as principled priors for inverse imaging. ICCV, 2023.
>
> [Chung&Kim 2023] H. Chung, J. Kim, M.T. Mccann, M.L. Klasky, and JC Ye. Diffusion Posterior Sampling for General Noisy Inverse Problems. ICLR, 2023.
>
> [EHT 2019] The EHT Collaboration et al. First M87 Event Horizon Telescope Results. IV. Imaging the Central Supermassive Black Hole. ApJL, 2019.

---

> > ### Comment · Reviewer_huK6 · 2023-08-12
> >
> > I appreciate the authors for their response.
> >
> > While going over the provided comments from other reviews, I have noticed a recurring concern regarding the computational cost linked with the non-amortized variational inference (VI) scheme. I can empathize with these concerns and believe that applying this approach to high-dimensional problems with forward operators that are expensive to evaluate could potentially lead to challenges in terms of scalability.
> >
> > However, I also acknowledge the authors' valid point about achieving reliable posterior approximations in scientific computing applications by explicitly integrating the forward operator into the inference scheme. Considering that existing preconditioned non-amortized variational inference techniques [1, 2]—readily applicable to the proposed method—offer a significant reduction in the computational costs associated with non-amortized variational inference while still providing dependable posterior estimates, I am less apprehensive about the incurred costs and would like to maintain the same score.
> >
> > I hope that other reviewers recognize that this technique has been devised while keeping in mind the intricacies of scientific computing applications, and furthermore, there are methods that enable scalable non-amortized variational inference.
> >
> > [1] A. Siahkoohi, G. Rizzuti, M. Louboutin, P. Witte, and F. J. Herrmann, “Preconditioned training of normalizing flows for variational inference in inverse problems,” in 3rd Symposium on Advances in Approximate Bayesian Inference, Jan. 2021.
> >
> > [2] A. Siahkoohi, G. Rizzuti, R. Orozco, and F. J. Herrmann, “Reliable amortized variational inference with physics-based latent distribution correction,” Geophysics, vol. 88, no. 3, R297–R322, Jan. 2023.

---

### Author Rebuttal · Authors · 2023-08-10

We thank all the reviewers for their encouraging feedback and for recognizing our “significant contribution to the field” (huK6): enabling large-scale Bayesian inference with a diffusion-model prior.

**Key contribution.** Our aim is to make score-based priors computationally feasible for inference of large images. As noted by all the reviewers, we achieve this goal with “effective gains in training time and memory footprints” (sVNv) and “a drastic reduction of computational time and memory cost while maintaining … very similar performance” (iehD) to the approach in [Feng 2023].

The approach of turning a diffusion model into a standalone prior brings benefits that are not offered by the diffusion-model-based sampling methods referenced by the reviewers ([Chung&Kim 2023] and [Trippe&Yim 2023]). These include:
* **Hyperparameter-free inference.** Unlike [Chung&Kim 2023] and similar methods, there are no measurement weights since we strictly follow the log-posterior formula.
* **Flexibility.** Users have freedom in choosing the optimization algorithm. In other methods like [Chung&Kim 2023] and [Trippe&Yim 2023], the sampling method is fixed. Although we demonstrate a variational-inference approach, our proposed surrogate score-based prior can be plugged into any optimization objective function that requires a differentiable log-probability.

[Feng 2023] has recently been accepted to ICCV based on these merits despite its computational limitations. Our work makes it possible to realize these same advantages with far-greater computational efficiency. For example, inference of an image with 64x64 pixels took over 5 days with the exact approach ([Feng 2023]) but just 45 minutes with ours (Fig. 2 in the manuscript).

**Key application areas.** Our approach is particularly useful for scientific and medical applications, where it is critical to understand the uncertainty in recovered features. While iehD noted that our method is less likely to hallucinate than [Chung&Kim 2023], [Jalal&Arvinte 2021], and [Song 2022], we disagree with their assumption that this is not worth the computational cost -- strong hallucinations unsupported by the measurements should not be tolerated in scientific analysis.

As Reviewers iehD and sVNv observe, the variational-inference approach we take requires fitting a distinct posterior for each set of measurements. However, it is common in scientific applications to be concerned with a single set of measurements and spend significant time on recovering the best-possible image from those measurements. One example is black-hole imaging, where the difficulty of obtaining measurements warrants a careful approach to image reconstruction.

In fact, since the submission, we have applied our method to black-hole imaging using the Event Horizon Telescope data published in [EHT 2019]. The efficiency of the proposed method made it possible to quickly iterate on our findings with this data. Our recent experience confirms the practical usefulness of the surrogate score-based prior, and we note that it is always possible to refine results with the slower, yet more exact, approach of [Feng 2023].

**Exactness of the posterior (iehD, sVNv).** Although our motivation to perform Bayesian inference leads us to minimize the upper-bound of a variational loss, we do not claim to sample from the exact posterior and will add language to the text to clarify this. Note that exact posterior sampling is rare, especially for high-dimensional images. Variational inference is itself an approximation of exact Bayesian inference, yet this technique has proven influential in many fields.


References:

[Feng 2023] B.T. Feng, J. Smith, M. Rubinstein, H. Chang, K.L. Bouman, and W.T. Freeman. Score-based diffusion models as principled priors for inverse imaging. ICCV, 2023.

[Chung&Kim 2023] H. Chung, J. Kim, M.T. Mccann, M.L. Klasky, and JC Ye. Diffusion Posterior Sampling for General Noisy Inverse Problems. ICLR, 2023.

[Trippe&Yim 2023] B.L. Trippe, J. Yim, D. Tischer, D. Baker, T. Broderick, R. Barzilay, and T.S. Jaakkola. Diffusion Probabilistic Modeling of Protein Backbones in 3D for the motif-scaffolding problem. ICLR, 2023.

[Jalal&Arvinte 2021] A. Jalal, M. Arvinte, G. Daras, E. Price, A.G. Dimakis, and J.I. Tamir. Robust compressed sensing MRI with deep generative priors. NeurIPS, 2021.

[Song 2022] Y. Song, L. Shen, L. Xing, and S. Ermon. Solving inverse problems in medical imaging with score-based generative models. ICLR, 2022.

[EHT 2019] The EHT Collaboration et al. First M87 Event Horizon Telescope Results. IV. Imaging the Central Supermassive Black Hole. ApJL, 2019.

---

### Author Response · Authors · 2023-08-20
**Results of ground-truth experiment proposed to iehD and sVNv**

We would like to provide results on a toy bimodal posterior in order to clarify the additional experiment that we proposed in our responses to iehD and sVNv. We performed an experiment on a 2D mixture-of-Gaussians with two Gaussian components (the prior was a bimodal mixture-of-Gaussians, and the forward model was linear, making the posterior a known bimodal posterior). This type of experiment allows us to carefully quantify accuracy w.r.t. a true posterior and assess performance over a reasonable space of hyperparameters for baseline methods.

As requested by iehD and sVNv, we compared to [Feng 2023] and [Chung&Kim 2023]. We also compared to [Jalal&Arvinte 2021], and [Song 2022]. [Feng 2023] uses the same variational-inference approach (DPI) as ours but with the exact score-based prior, which is computationally expensive. The other three baselines are all diffusion-based and involve hyperparameters for the measurement weight. We used the same score function for each method. In order to compare the probability density function (PDF) across methods, we fit each method’s posterior samples to a two-component Gaussian mixture model (GMM).

***

**KL divergence.** The KL divergence values are as follows (lower is better). For baselines with hyperparameters, we report the median and the best KL divergence (according to ground-truth) across the hyperparameter search.

Ours (DPI + surrogate score-based prior): **0.037**

Feng 2023 (DPI + exact score-based prior): **0.030**

Song 2022: 0.95 (median), 0.12 (best)

Jalal&Arvinte 2021: 0.96 (median), 0.10 (best)

Chung&Kim 2023: 0.55 (median), 0.067 (best)

***

**Accuracy of multimodal characterization.** All methods recovered two modes in the posterior, but only ours and [Feng 2023] correctly estimated that one mode was stronger than the other. Below are the weights of the two components of the fitted GMM (closer to “True” is better). Note that [Song 2022], [Jalal&Arvinte 2021], and [Chung&Kim 2023] incorrectly place equal weight on each mode.

True: [0.70, 0.30]

Ours (DPI + surrogate score-based prior): [0.61, 0.39]

Feng 2023 (DPI + exact score-based prior): [0.70, 0.30]

Song 2022: [0.50, 0.50] (best)

Jalal&Arvinte 2021: [0.49, 0.51] (best)

Chung&Kim 2023: [0.53, 0.47] (best)

***

Even with a simple 2D case, the benefits of our method are clear:

* **More-accurate posterior without hyperparameter-tuning.** [Song 2022], [Jalal&Arvinte 2021], and [Chung&Kim 2023] only achieve a good KL divergence when their hyperparameters can be tuned according to a known ground-truth. We emphasize that real-world applications do not allow for a ground-truth, making this type of hyperparameter optimization impossible with baseline methods. *Even with the best hyperparameter setting, these baselines do not accurately recover the multimodal posterior.*
* **Comparable performance to [Feng 2023] for much less cost.** The KL divergence of our method rivals that of [Feng 2023]. The benefit of ours is computational efficiency: [Feng 2023] took 130 ms per DPI optimization step, while ours took 22 ms (we ran each algorithm for 12000 steps total). We highlight that our 6x runtime improvement is for 2D data, and as shown in Fig. 2 of the submitted manuscript, the improvement becomes much more significant as the data dimensionality increases.

We have already prepared a figure illustrating our results and will include it in the revised manuscript.

***

References:

[Feng 2023] B.T. Feng, J. Smith, M. Rubinstein, H. Chang, K.L. Bouman, and W.T. Freeman. Score-based diffusion models as principled priors for inverse imaging. ICCV, 2023.

[Chung&Kim 2023] H. Chung, J. Kim, M.T. Mccann, M.L. Klasky, and JC Ye. Diffusion Posterior Sampling for General Noisy Inverse Problems. ICLR, 2023.

[Jalal&Arvinte 2021] A. Jalal, M. Arvinte, G. Daras, E. Price, A.G. Dimakis, and J.I. Tamir. Robust compressed sensing MRI with deep generative priors. NeurIPS, 2021.

[Song 2022] Y. Song, L. Shen, L. Xing, and S. Ermon. Solving inverse problems in medical imaging with score-based generative models. ICLR, 2022.

---

### Decision · Program_Chairs · 2023-09-21

**Decision:**

Reject

**Comment:**

The paper presents a method for performing variational inference on models with diffusion based priors. The paper builds on a method proposed in citation [10] from arxiv, and the contribution is intended to be the improved time and memory requirements of the proposed approximation. The quantitative and qualitative comparisons are mostly limited to a comparison with [10] and the performance of this particular inverse method for image denoising isn’t considered. While one reviewer was very enthusiastic, this person acknowledged this shortcoming and the still-high computational cost. The other two reviewers leaned towards rejection for these reasons. While the paper may address a concern with [10], limiting the scope of the paper to the short fix (Sec. 4) of this problem was unfortunately not a significant enough contribution for this year’s NeurIPS.